# Circulating microRNAs as Early Biomarkers of Colon Cancer: A Nested Case-Control Study Within a Prospective Cohort

**DOI:** 10.3390/ijms26167893

**Published:** 2025-08-15

**Authors:** Lisa Padroni, Giorgia Marmiroli, Laura De Marco, Valentina Fiano, Lucia Dansero, Saverio Caini, Giovanna Masala, Luca Manfredi, Lorenzo Milani, Fulvio Ricceri, Carlotta Sacerdote

**Affiliations:** 1Department of Clinical and Biological Sciences, Centre for Biostatistics, Epidemiology, and Public Health (C-BEPH), University of Turin, 10043 Turin, Italy; lisa.padroni@unito.it (L.P.); lucia.dansero@unito.it (L.D.); luca.manfredi@unito.it (L.M.); lorenzo.milani@unito.it (L.M.); fulvio.ricceri@unito.it (F.R.); 2Unit of Cancer Epidemiology, Department of Medical Sciences, University of Turin, 10126 Turin, Italy; giorgia.marmiroli@unito.it (G.M.); valentina.fiano@unito.it (V.F.); 3Unit of Cancer Epidemiology, Città Della Salute e Della Scienza University-Hospital and Center for Cancer Prevention (CPO), Via Santena 7, 10126 Turin, Italy; laura.demarco@cpo.it; 4Cancer Risk Factors and Lifestyle Epidemiology Unit, Institute for Cancer Research, Prevention and Clinical Network (ISPRO), 50139 Florence, Italy; s.caini@ispro.toscana.it; 5Clinical Epidemiology Unit, Institute for Cancer Research, Prevention and Clinical Network (ISPRO), 50139 Florence, Italy; g.masala@ispro.toscana.it; 6Department of Health Sciences, University of Eastern Piedmont, Via Solaroli 17, 28100 Novara, Italy; 7Unit of Epidemiology, Local Health Unit of Novara, Viale Roma, 7, 28100 Novara, Italy

**Keywords:** colon cancer, microRNA, nested case-control study

## Abstract

Circulating microRNAs (miRNAs) have emerged as non-invasive biomarkers that may be associated with cancer risk, but their role in the development of colon cancer is still not well understood. We conducted a nested case-control study within the EPIC-Italy cohort to investigate the association between pre-diagnostic serum levels of eight candidate miRNAs (Let7, Mir21, Mir155, Mir181, Mir222, Mir145, Mir92, and Mir20) and subsequent colon cancer occurrence. A total of 104 incident colon cancer cases were matched to 104 controls by center, sex, age, recruitment date, and vital status. miRNA expression was quantified using RT-qPCR and normalized to Mir484. Logistic regression models were applied to estimate odds ratios, 95% confidence intervals, and *p*-values, adjusting for age at recruitment, smoking status, body mass index, physical activity, adherence to a Mediterranean diet, and socioeconomic position. Elevated expression of Let7 (OR = 0.91; 95% CI: 0.84–1.00; *p* = 0.04) was associated with slightly lower odds of colon cancer in unadjusted models. Mir21 and Mir222 showed borderline associations (*p* = 0.07 and *p* = 0.09, respectively), but these did not remain significant after Bonferroni correction. This result was consistent in the multivariate logistic model: higher levels of Let7 (OR = 0.91; 95% CI: 0.82–1.00; *p* = 0.06) and Mir222 (OR = 0.75; 95% CI: 0.57–1.00; *p* = 0.05) are suggestive of an association with lower odds of colon cancer. Our findings highlight the challenges of using circulating miRNAs as very early biomarkers, particularly when samples are collected nearly a decade before diagnosis. Future studies with larger sample sizes, serial blood collections, and integration with inflammatory and immune markers will be crucial to clarify the temporal dynamics of circulating miRNA alterations and their potential role in risk-adapted screening strategies.

## 1. Introduction

MicroRNAs (miRNAs) are small non-coding RNA molecules, typically 18–25 nucleotides in length, that play a key role in the post-transcriptional regulation of gene expression. By binding to complementary sequences on target messenger RNAs (mRNAs), miRNAs can promote mRNA degradation or inhibit translation, thereby modulating protein synthesis. [1,2]

miRNAs have been suggested to be involved in the regulation of a wide range of physiological and pathological processes, including development, immune responses, inflammation, and cellular homeostasis. [3] Given their broad regulatory functions, it has been hypothesized that miRNAs may also contribute to cancer-related processes, including cell proliferation, apoptosis, angiogenesis, and metastasis. [4]

Although the causal role of specific miRNAs in cancer initiation and progression remains controversial, accumulating evidence points to their potential as minimally invasive biomarkers for early cancer detection. [4] Among various malignancies, colon cancer (CC) appears to exhibit relatively stable and reproducible miRNA expression signatures, which may reflect early molecular changes during carcinogenesis. [5]

In this study, we aim to investigate the association between a selected panel of miRNAs and the subsequent development of colorectal cancer. We specifically focus on miRNA expression profiles measured several years before the clinical onset of the disease, based on a nested case-control design within a large prospective cohort.

For our analysis panel, we selected eight circulating miRNAs—Let7, Mir21, Mir155, Mir181, Mir222, Mir145, Mir92, and Mir20 [6,7,8,9,10,11,12,13,14,15,16,17,18,19]. These candidates were chosen based on previous evidence linking them to CC across different stages of tumorigenesis, from early initiation and growth to metastatic dissemination [20,21]. To provide the biological rationale for this selection and a concise overview of their relevance, Table 1 summarizes the established roles of these miRNAs in CC, their main molecular targets and pathways, and key supporting references.

The Let7 family of miRNAs has been studied in large prospective cohorts such as the Nurses’ Health Study and Health Professionals Follow-up Study, which analyzed tumor tissue from hundreds of CC patients [7]. These studies identified that elevated tumor Let7 expression correlates with reduced infiltration of cytotoxic and memory T cells (CD3+ and CD45RO+), linking Let7 to immune suppression within the tumor microenvironment and worse cancer-specific survival. This suggests a role for Let7 in mediating immune evasion mechanisms rather than directly driving tumor cell proliferation or metastasis [7].

Mir21 is among the most extensively studied miRNAs with diagnostic and prognostic relevance in CC. Multiple nested case-control studies and cohort studies have demonstrated plasma Mir21 levels to be significantly elevated in patients before CC diagnosis compared to controls [8,22]. Its differential expression promotes tumor cell proliferation and metastasis by targeting tumor suppressors such as PDCD4 and PTEN and regulating inflammatory pathways, including NF-κB [8].

Mir155 contributes to CC progression through enhancing cell migration and invasion, driven by its modulation of β-catenin and RhoA signaling [11]. Beyond tumor cell-autonomous effects, Mir155 has a key role in shaping the inflammatory milieu by regulating immune cells like myeloid-derived suppressor cells and tumor-associated macrophages, accentuating its involvement in tumor-promoting inflammation and metastasis [11].

Mir181 has been shown in functional in vitro and in vivo studies to stimulate CC cell proliferation and invasiveness by targeting the tumor suppressor CYLD and activating NF-κB signaling, an important mediator of inflammation, underscoring the convergence of oncogenic signaling and inflammatory pathways [12].

Mir222, characterized in models of inflammation-associated CC, including colitis-induced carcinogenesis, promotes tumor progression by enhancing inflammatory signaling and oxidative stress in the intestinal mucosa [13]. Suppression of Mir222 ameliorates colitis and reduces inflammation-driven tumor growth, highlighting a direct mechanistic link between intestinal inflammation and CC progression mediated by Mir222 [14].

Mir145 typically functions as a tumor suppressor in CC, frequently downregulated in tumor tissues. It impacts tumor proliferation and metastatic spread through negative regulation of key oncogenes [16]. Though less directly linked to inflammation, Mir145 downregulation contributes to an environment permissive of oncogenic and inflammatory signaling, indirectly influencing tumor-associated inflammation [16].

Mir92a, studied in nested case-control cohorts with prediagnostic plasma samples, is elevated in early CC and advanced adenomas compared to healthy individuals. It participates in tumor induction and progression by targeting multiple tumor-suppressive pathways, and its expression may reflect subclinical inflammatory states associated with colorectal neoplasia [17].

Lastly, Mir20a has been identified in cohort analyses as differentially regulated in CC patient blood and tissues. It promotes tumor proliferation and progression by regulating chemokines such as CXCL8, which also contribute to the recruitment of inflammatory cells in the tumor microenvironment, showcasing a dual role in oncogenic and inflammatory regulation [19].

Although tissue-based studies have consistently implicated these miRNAs in colon cancer development, their detectability in circulation many years before diagnosis remains controversial [23]. Associations observed in tumor tissue are generally robust and reproducible, whereas studies on circulating miRNAs have produced highly variable results, even when comparing plasma and serum within similar clinical settings [23]. Moreover, only a limited number of studies have analyzed pre-diagnostic blood samples, and very few have investigated samples collected several years prior to cancer onset, when circulating miRNA alterations are likely to be subtle and difficult to detect [24]. Actively or passively, miRNAs released from tumor cells—e.g., through apoptosis or necrosis—may be undetectable or negligible during the earliest, microscopic, or pre-malignant phases [25].

In summary, the selected miRNAs could clarify the multifaceted interplay between oncogenic signaling, immune regulation, and inflammation in colorectal cancer pathogenesis. Their detection in circulating blood prior to clinical diagnosis could underscore their potential as early biomarkers, while future functional studies will provide mechanistic insight into how miRNAs modulate tumor biology and inflammatory states.

## 2. Results

The study population included 208 participants, 104 incident colon cancer cases, and 104 healthy controls. The average follow-up duration until the occurrence of colon cancer among cases was 9.6 years (min 1.05 years; max 18.32 years). The main socio-demographic characteristics of the study population were comparable between cases and controls. The mean age at the recruitment is slightly higher among cases than controls (55.8 versus 54.3 years, *p* = 0.09). No statistically significant differences have been observed between the two groups for age, smoking status, body mass index, physical activity score, Mediterranean diet index, and relative index of inequality (Table 2).

The levels of eight different miRNAs were analyzed (Let7, Mir21, Mir155, and Mir181, Mir222, Mir145, Mir92, Mir20). The Wilcoxon test did not show any statistically significant differences in miRNA expression levels between cases and controls. However, two miRNAs (Let7 and Mir222) showed results close to statistical significance, suggesting that further investigation may be required. (Figure 1)

We have included in the Appendix A boxplots for the eight analyzed miRNAs, showing their expression levels normalized to miR-484 without log transformation (Appendix A).

When comparing mean expression levels of the eight analyzed miRNAs between incident colon cancer cases and healthy controls, we observed that almost all miRNAs (Let7, Mir21, Mir155, Mir181, Mir222, Mir145, and Mir20) were upregulated in controls, while only one miRNA (Mir92) showed a higher mean expression level in cases compared to controls. Specifically, the fold change values ranged from 0.21 to 1.16 (Table 3).

Subsequently, a univariate regression was performed for each miRNA for cases and controls. The analysis highlighted that Let7 was significantly associated with colon cancer (*p* = 0.04), while Mir21 (*p* = 0.07) and Mir222 (*p* = 0.09) suggested a slight association, consistent with what emerged from the Wilcoxon test. To control for the risk due to multiple hypothesis testing, a Bonferroni correction was applied. None of the miRNAs were found to be significantly associated with cancer after adjustment (Table 4).

A multivariate logistic regression was performed to evaluate the association between the expression of the analyzed miRNAs and colon cancer, considering socio-demographic confounding variables: age at recruitment, smoking status, BMI, level of physical activity, adherence to the Mediterranean diet, and relative index of inequality (RII). The Mir222 (OR = 0.75, IC 95%: 0.57–1.00, *p* = 0.05) and Let7 (OR = 0.91, IC 95%: 0.82–1.00, *p* = 0.06) showed an association with colon cancer status. Still, none of the miRNAs showed a significant association with colon cancer status in this adjusted model after Bonferroni correction for multiple comparisons (Table 5).

## 3. Discussion

This nested case-control study, conducted within a prospective epidemiological cohort, aimed to validate a panel of eight microRNAs that had been previously identified as differentially expressed in the serum of colorectal cancer cases and controls from Caucasian populations. The primary objective of this validation effort was to address and overcome some of the methodological limitations commonly observed in microRNA case-control studies published to date.

A key strength of this study lies in its prospective design. By using pre-diagnostic blood samples collected from individuals before the onset of colon cancer, we were able to minimize the risk of reverse causality—a frequent concern in retrospective studies. This aspect is particularly important, as it allows for a more accurate assessment of the potential role of these microRNAs as early biomarkers for cancer detection.

Another notable advantage of our approach is the implementation of multivariable analyses. We adjusted for a comprehensive set of potential confounders, mainly lifestyle risk factors, which is not routinely performed in similar studies investigating circulating microRNAs. Many previous studies on miRNA and cancer risk have relied mainly on univariate analyses. Anyway, it is certainly essential to perform multivariate analyses that take potential confounders into account if we aim to investigate a possible causal association. Such analyses are also important when searching for early biomarkers of disease. Finally, none of the associations between miRNAs and colon cancer risk remained significant after correction for multiple testing. To ensure methodological rigor, we applied the Bonferroni correction, which is well-known to be highly conservative.

In addition, we took care to account for multiple comparisons in our statistical analyses, thereby reducing the likelihood of false-positive findings. This methodological precaution is particularly important in biomarker studies involving multiple molecular targets, where the risk of type I error is inherently elevated.

This analytical strategy strengthens the validity of our findings by accounting for alternative explanations of the observed associations.

Nevertheless, when blood samples are collected before cancer onset, the circulating miRNAs detected are likely to reflect indirect systemic changes—such as immune or inflammatory responses—rather than direct tumor secretion [25]. The use of pre-diagnostic samples collected nearly a decade before diagnosis increases confidence against reverse causality but also highlights a temporal challenge: molecular markers may not reach detectability thresholds so far in advance, especially those linked primarily to later carcinogenic stages or systemic inflammatory responses [24].

Inflammation-induced miRNAs could emerge earlier as paraneoplastic signals. Chronic inflammation, metabolic stress, or immune dysregulation—arising from nascent tumors or field carcinogenesis—may systemically alter miRNA profiles [24].

Among the eight miRNAs analyzed, Let7, Mir21, and Mir222 emerged as the most promising candidates, showing consistent differential regulation in cases compared to controls, with borderline significance in univariate analyses before multiple testing correction. These findings are in line with previous evidence suggesting their involvement in colorectal carcinogenesis through pathways related to cell proliferation, apoptosis, and invasion [21]. When we adjusted for major potential confounders, Let7, which was statistically significant in the univariate models, lost its significance. Conversely, Mir222, which did not reach significance in the univariate analysis, showed a borderline statistical significance after multivariable adjustment, suggesting that confounding factors may have masked its association in the unadjusted model.

These findings align with their well-established roles in tumor biology, particularly regarding proliferation, invasion, and modulation of the immune and inflammatory environment [7,14,26].

Given the function of Let7 as a tumor suppressor often downregulated in precancerous lesions, its early dysregulation may initially be confined to local tissue changes, which might not yet translate into measurable alterations in circulating levels. Circulating miRNAs like Let7 may only become detectable when the neoplasm has sufficiently progressed to release signals into the bloodstream or when systemic inflammatory processes become more pronounced. This temporal delay in appearance could explain the modest associations observed several years before diagnosis [7,25].

The oncogenic role of Mir21 is strongly linked to advanced cancer phenotypes, including enhanced cell survival and inflammation. Its presence in circulation years prior to clinical detection may therefore be subtle, reflecting a developing systemic pro-inflammatory state rather than early tumor burden. The statistically consistent but modest differential expression observed supports this gradual emergence as disease progresses [26,27].

The association of Mir222 with metastatic disease and inflammatory bowel pathology suggests it may act as a bridge between chronic intestinal inflammation and tumorigenesis [14]. Its modulation in preclinical inflammation-driven tumor models reinforces the hypothesis that changes in circulating Mir222 could indicate a pro-tumorigenic inflammatory milieu, detectable even prior to overt malignancy. Notably, its borderline significance after adjusting for confounding factors implies that specific host or environmental variables may influence its circulating levels [14].

In contrast, several expected candidates, including Mir155, Mir181, Mir145, Mir92a, and Mir20a, did not show significant associations in this early pre-diagnosis window but are nonetheless biologically expected to be involved based on their established roles in the earliest stages of tumor growth and the creation of a proinflammatory microenvironment [28,29].

Moreover, chronic inflammation of the intestinal mucosa, a well-known driver of colorectal carcinogenesis, is regulated in part by specific miRNAs that influence immune cell activity and cytokine signaling. miRNAs such as Mir155, Mir181, Mir92a, and Mir20a have been implicated in sustaining a tumor-promoting inflammatory milieu, with roles in immune regulation, macrophage polarization, and modulation of chemokine expression [30,31]. These inflammation-related miRNAs may be altered prior to tumor detectability, reflecting field cancerization effects or systemic immune dysregulation [32]. However, miRNA alterations in circulation during these very early phases are often subtle and transient, making their detection in pre-diagnostic blood samples challenging. This limitation likely explains the lack of significant associations in our study, where samples were collected many years before clinical diagnosis [33].

In conclusion, a major limitation of our study is the relatively small sample size, which does not allow us to confirm whether the suggestive associations we observed are true signals. However, the study design and the application of comprehensive statistical adjustments are essential to minimize the risk of bias, and we hope these methodological standards will be followed by future studies with larger sample sizes. Moreover, the a priori selection of candidate miRNAs, although based on the existing literature and biological plausibility, may have introduced selection bias and limited the discovery of novel biomarkers. An unbiased, hypothesis-free approach covering the entire miRNome would be more appropriate for future comprehensive biomarker discovery.

Despite these limitations, our findings provide useful preliminary insights and may serve as a foundation for future studies with larger sample sizes and broader analytical strategies.

Our findings highlight the intrinsic challenges of using circulating microRNAs as very early biomarkers for colorectal cancer. Most blood-based studies have analyzed samples collected within one to two years before diagnosis, whereas in our cohort the average interval was nearly ten years, when tumors were likely microscopic or pre-malignant. At such early stages, tumor-derived miRNAs may remain below the detection threshold while early dysregulation—such as the silencing of tumor-suppressor miRNAs like Let7—likely occurs locally in the colonic epithelium without yet reaching the circulation.

Future research should aim to validate these associations in larger and pooled cohorts and, ideally, to incorporate serial blood sampling that captures the dynamic evolution of miRNA expression as individuals approach diagnosis. Tracking temporal changes would clarify whether certain miRNAs gradually increase or decrease in preclinical phases, thereby improving their predictive value. Coupled with functional studies to distinguish causal from merely reflective signals, this approach could ultimately identify the critical window in which circulating miRNAs transition from undetectable to clinically informative, paving the way for their integration into risk-adapted screening and early detection strategies.

## 4. Materials and Methods

### 4.1. Study Design and Population

This study is based on a nested case-control design within the European Prospective Investigation into Cancer and Nutrition (EPIC)-Italy cohort (Florence and Turin centers). EPIC-Italy is part of the larger EPIC study, a multicenter prospective cohort involving over half a million participants across 10 European countries. In Italy, approximately 47,000 adult volunteers were recruited between 1993 and 1998 from five centers (Florence, Turin, Naples, Ragusa, and Varese) [34]. At recruitment, participants provided detailed dietary and lifestyle information through standardized questionnaires and underwent anthropometric measurements and blood sampling. Importantly, blood samples were collected at baseline, several years prior to any cancer diagnosis, allowing for prospective biomarker analyses.

### 4.2. Case and Control Selection

All incident cases of colon cancer diagnosed after recruitment were identified through cancer registries and active follow-up, with cancer follow-up extending from the date of enrollment until 2010. We excluded individuals with a history of prevalent cancer. Only cases of histologically confirmed, malignant adenocarcinoma colon cancer were included. From each center, 65 cases were selected and individually matched to healthy controls using incidence density sampling. Matching criteria included sex, age at recruitment (±2.5 years), center, date of recruitment (±6 months), and use of hormone replacement therapy (for women). Only participants with complete baseline dietary and lifestyle questionnaires were eligible for inclusion.

### 4.3. Selection of miRNAs

The miRNAs analyzed in this study—Let7, Mir21, Mir155, Mir181, Mir222, Mir145, Mir92, and Mir20—were selected based on a systematic literature review. [20] We prioritized those miRNAs that showed the strongest associations with case-control status in population-based studies conducted in Caucasian subjects and in which miRNA expression had been measured in serum samples.

### 4.4. Laboratory Methods

Total RNA was extracted from serum of selected patients using the miRNeasy Serum/Plasma Advanced Kit (Qiagen, Hilden, Germany) according to the manufacturer’s protocol and stored at −80 °C.

Reverse transcription was carried out using the miRCURY LNA RT Kit (Qiagen) with the following modification to the manufacturer’s protocol: 3 ul of RT SYBR Green Reaction Buffer, 1.5 μL of miRCURY RT Enzyme Mix, 0.75 μL of UniSp6 RNA Spike, 3 μL of RNA sample, and 6.75 μL of H_2_O reaching a final volume of 15 μL. The thermal cycling was conducted according to the manufacturer’s instructions.

QX200 Droplet Digital PCR System (Biorad, Laboratories, Hercules, CA, USA) was used to measure the expression level of Mir21, Mir155, Mir222, Mir145, Mir181, Mir92, and Let7. A PCR mix with a final volume of 21.5 μL was prepared containing 10 μL of QX200 EvaGreen SuperMix (Biorad), 2 μL of miRCURY LNA miRNA PCR Assay (Qiagen) for each selected miRNA, 8 μL of di H_2_O and 1.5 μL of di cDNA sample. Thermal cycling conditions were as follows: 95 °C for 5 min, then 40 cycles of 95 °C for 30 s and 58 °C for 1 min, and three final steps at 4 °C for 5 min, 90 °C for 5 min, and a 4 °C infinite hold.

Data analysis was performed using the QX Manager Software 2.0 Standard Edition, and the results were considered valid if the number of counted droplets reached at least 10,000. The miRNA concentration is calculated as miRNA copies/μL normalized to the Mir484.

### 4.5. Statistical Analysis

A descriptive exploration of the expression of miRNAs, in cases and controls, was performed, calculating for each miRNA the mean of normalized values and the ratio between means in cases and controls (fold change).

Differences between groups were evaluated using the Wilcoxon rank-sum test to compare expression distributions between cases and controls in each miRNA analyzed.

Descriptive analysis was illustrated using boxplots. To account for multiple comparisons, *p*-values were adjusted using the Bonferroni correction method.

Subsequently, a univariate logistic regression was performed for each miRNA in order to estimate the crude association between disease status (case/control) and miRNA expression levels.

To conclude, a multivariate logistic regression model was applied for each miRNA, including age at recruitment, smoking status, Body Mass Index (BMI), level of physical activity at recruitment, a score of the adherence to the Mediterranean diet, and the relative index of inequality. For each miRNA, the odds ratio, 95% confidence interval, and *p*-value were reported.

The BMI has been categorized as normal weight where the BMI value was less than 25, overweight if the BMI value was between 25 and 30, and obese if the BMI value was over 30.

Age at recruitment was included in the models as a continuous variable.

Smoking status was categorized as current smokers, former smokers, and never smokers.

Physical activity was measured using a validated index that combines occupational, household, and recreational activities, expressed in metabolic equivalent task-hours per week (MET-h/week). Based on sex-specific quartiles, participants were classified into four categories: inactive, moderately inactive, moderately active, and active [35].

Adherence to the Mediterranean diet was assessed using the Mediterranean Diet Score (MDS), which evaluates intake of nine dietary components: vegetables, legumes, fruits and nuts, cereals, fish, meat, dairy products, alcohol, and the ratio of monounsaturated to saturated fats. Each component contributes 0 or 1 point to the total score depending on sex-specific median intake, with a total score ranging from 0 (minimal adherence) to 9 (maximal adherence). The MDS has been previously validated within the EPIC cohort as a proxy for adherence to a Mediterranean dietary pattern [36].

Socioeconomic position was assessed using the Relative Index of Inequality (RII), calculated based on participants’ sex, age group at recruitment, and educational level. The RII is a summary measure of relative socioeconomic inequality that accounts for the distribution of education within the population. It was categorized into three levels: high (RII = 1), intermediate (RII = 2), and low socioeconomic position (RII = 3), with lower values indicating higher socioeconomic status [37].

All analyses were carried out using the R Studio software (R version 4.4.3) [38].

## Figures and Tables

**Figure 1 ijms-26-07893-f001:**
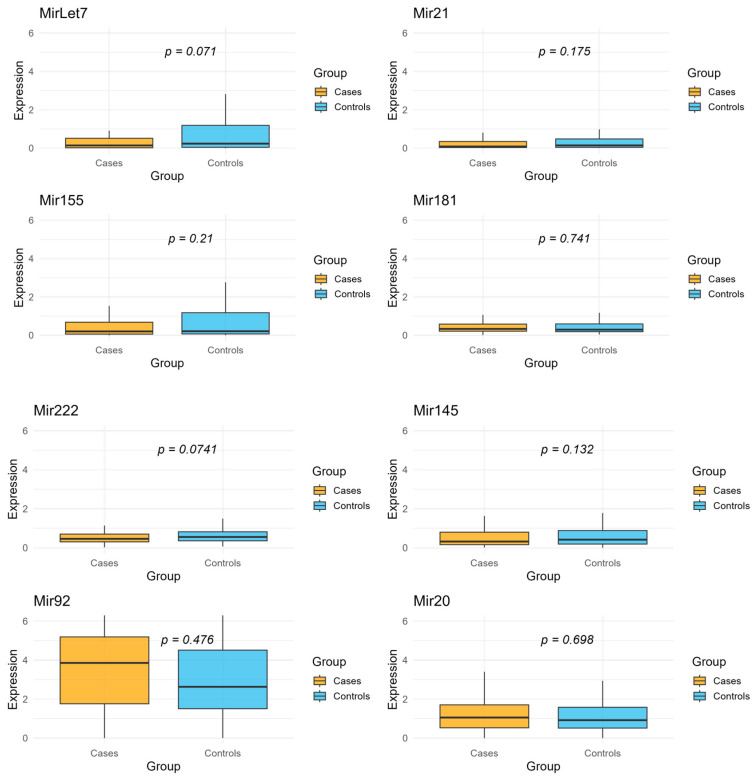
Expression levels of normalized and log-transformed selected miRNAs (Let7, Mir21, Mir155, Mir181, Mir222, Mir145, and Mir20) in incident colon cancer cases and healthy controls. Box plots display the median values (horizontal line) with interquartile ranges (IQR; boxes) and whiskers indicating the data range. *p*-values are from Wilcoxon rank-sum tests comparing the distribution of expression levels between cases and controls for each miRNA.

**Table 1 ijms-26-07893-t001:** Overview of the selected circulating miRNAs analyzed in this study, including their reported roles in colorectal cancer, key molecular targets and pathways, and main supporting references.

miRNA	Role in CC	Key Targets/Pathways	References
Let7	Tumor suppressor; promotes differentiation and inhibits proliferation. Suppressed by LIN28A/B.	LIN28A/B; regulates cell differentiation; anti-proliferative	[6,7]
Mir21	Oncogenic; promotes proliferation, invasion, inflammation, and therapy resistance.	PDCD4, PTEN; Wnt/β-catenin and AKT pathways	[8,9]
Mir155	Oncogenic; promotes migration, invasion, EMT, and inflammation.	AXIN1, TCF4; Wnt/β-catenin; CAF/TAM exosomal signaling	[10,11]
Mir181	Promotes tumor progression and therapy resistance; involved in Wnt signaling.	Wnt/β-catenin; CYLD, modulated by lncRNAs	[12]
Mir222	Oncogenic; promotes migration, invasion, metastasis, and immune escape.	MST3, SPINT1, ADAM-17, ATF3; chemoresistance	[13,14]
Mir145	Tumor suppressor; downregulated in early CC; promotes differentiation.	OCT4, SOX2, EGFR, RREB1; Ras-MAPK and Wnt regulation	[15,16]
Mir92a	Oncogenic; member of Mir17~92 cluster; promotes proliferation and angiogenesis.	DKK3, SOCS3, NF2, KLF4	[17,18]
Mir 20a	Oncogenic; part of Mir17~92 cluster; promotes proliferation and inhibits apoptosis.	ATG5, FIP200, CXCL8, PDCD4, MICA, FOXJ2, PTEN, SMAD4	[19]

**Table 2 ijms-26-07893-t002:** Baseline socio-demographic characteristics of the study population, including 208 participants (104 incident colon cancer cases and 104 healthy controls). Values are presented as mean ± standard deviation (SD) or number (percentage). *p*-values are based on the Wilcoxon rank sum test, Pearson’s Chi-squared test, or Fisher’s exact test, as appropriate.

Characteristics	Cases N = 104	Controls N = 104	*p*-Value
**Age (mean ± SD)**	55.8 ± 6.0	54.5 ± 6.5	0.09
**Center**			>0.90
Florence	53 (51%)	53 (51%)	
Turin	51 (49%)	51 (49%)	
**Smoking Status**			0.20
Never smokers	46 (44%)	47 (45%)	
Former smokers	38 (37%)	28 (27%)	
Smokers	20 (19%)	29 (28%)	
**Body Mass Index**			0.70
Normal weight (BMI < 25)	31 (31%)	37 (37%)	
Overweight (25 < BMI < 30)			
55 (54%)	51 (50%)		
Obese (BMI > 30)	15 (15%)	13 (13%)	
**Physical Activity**			0.50
Active	7 (6.7%)	12 (12%)	
Moderately active	38 (37%)	41 (39%)	
Moderately inactive	40 (38%)	36 (35%)	
Inactive	19 (18%)	15 (14%)	
**Mediterranean Score**			>0.90
0–2	26 (25.5%)	23 (22.6%)	
3–5	60 (57.0%)	59 (56.0%)	
6–10	18 (17.5%)	22 (21.4%)	
**Socioeconomic Position**			0.80
High	35 (35%)	30 (30%)	
Medium	27 (27%)	29 (29%)	
Low	38 (38%)	41 (41%)	

**Table 3 ijms-26-07893-t003:** Fold change in cases and controls for the panel of eight selected miRNAs. Mean expression levels were calculated for incident colon cancer cases and healthy controls, and fold change was computed as the ratio of mean expression in cases to mean expression in controls.

miRNA	Mean Expression in Cases	Mean Expression in Controls	Fold Change
Let7	0.68	2.63	0.26
Mir21	0.41	1.96	0.21
Mir155	0.56	1.50	0.37
Mir181	0.47	0.87	0.54
Mir222	0.57	1.52	0.38
Mir145	0.63	1.73	0.36
Mir92	3.91	3.36	1.16
Mir20	1.37	2.53	0.54

**Table 4 ijms-26-07893-t004:** Univariate logistic regression odds ratios, 95% confidence intervals, and corresponding *p*-values for the association between circulating levels of eight selected miRNAs and the risk of incident colon cancer. *p*-values adjusted for multiple comparisons were calculated using the Bonferroni correction.

miRNA	OR	Low IC 95%	High IC 95%	*p*-Value	Bonferroni Adjusted *p*-Value
Let7	0.91	0.84	1.00	0.04	0.33
Mir21	0.90	0.81	1.01	0.07	0.56
Mir155	0.92	0.83	1.02	0.10	0.83
Mir181	0.96	0.77	1.19	0.71	1.00
Mir222	0.82	0.64	1.03	0.09	0.74
Mir145	0.89	0.76	1.04	0.14	1.00
Mir92	0.97	0.85	1.12	0.68	1.00
Mir20	1.00	0.87	1.14	0.95	1.00

**Table 5 ijms-26-07893-t005:** Multivariate logistic regression odds ratios, 95% confidence intervals, and corresponding *p*-values for the association between circulating levels of eight selected miRNAs and the risk of incident colon cancer. *p*-values adjusted for multiple comparisons were calculated using the Bonferroni correction.

miRNA	OR	Low IC 95%	High IC 95%	*p*-Value	Bonferroni Adjusted *p*-Value
Let7	0.91	0.82	1.00	0.06	0.48
Mir21	0.92	0.81	1.04	0.18	1.00
Mir155	0.93	0.83	1.04	0.20	1.00
Mir181	0.97	0.75	1.25	0.83	1.00
Mir222	0.75	0.57	1.00	0.05	0.40
Mir145	0.88	0.73	1.06	0.17	1.00
Mir92	0.97	0.82	1.14	0.71	1.00
Mir20	1.01	0.87	1.18	0.86	1.00

## Data Availability

The data that support the findings of this study are available from the corresponding author upon reasonable request.

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
