# Peer review of "Circulating microRNAs as Early Biomarkers of Colon Cancer: A Nested Case-Control Study Within a Prospective Cohort"

_ijms, 2025, doi:10.3390/ijms26167893_

Round 1

Reviewer 1 Report

Comments and Suggestions for Authors

This study provides valuable insights into the potential role of circulating miRNAs as early biomarkers for colon cancer.

The selected miRNAs (Let-7, miR-21, miR-155, miR-222, miR-145, miR-181, miR-92, and miR-20) were prioritized based on their consistent dysregulation in CRC tissue and circulation across independent cohorts.

However, conflicting reports exist regarding their specificity in pre-diagnostic blood, as miRNAs (e.g., miR-21) also elevate in inflammatory conditions (e.g., colitis).

In this regard, this study evaluates whether these miRNAs reflect early CRC-specific changes or generalized risk-associated states.

However, the results also raise important questions about the timing of sample collection and study design: the timing of sample collection is critical, and the study’s design, while rigorous, may have missed miRNA signals that emerge closer to diagnosis. Addressing this could refine the potential of miRNAs as early detection tools.

Here are my thoughts and suggestions:

KEY OBSERVATIONS:

Most "early" miRNA studies in blood are still conducted within 1–2 years of diagnosis, not decades prior.

While tissue-based studies implicate these miRNAs in CRC development, their detectability in circulation years before diagnosis remains controversial. If blood samples were collected years before cancer diagnosis, the detected miRNAs likely reflect indirect systemic changes (e.g., immune/inflammatory responses) rather than direct tumor secretion. Tumor-derived miRNAs may be diluted by systemic noise until late stages. Actively or passively, tumor-released miRNAs (cell death) may be undetectable or negligible in very early stages when the tumor is microscopic or pre-malignant. Early miRNA changes may be tissue-specific but not yet detectable in blood. Dysregulation of tumor suppressor miRNAs like let7 (silenced in early adenomas) might occur locally in the colonic epithelium long before secretion into circulation reaches detectable levels. Circulating miRNAs could become measurable only after the tumor reaches a certain size, invades vasculature, or triggers systemic inflammation. This is particularly sustainable for miR-21, the most frequently upregulated oncomiR in CRC, associated with advanced-stage disease and poor prognosis.

Conversely, inflammation-driven miRNAs could emerge earlier as paraneoplastic signals: chronic inflammation, metabolic stress, or immune dysregulation (e.g., from nascent tumors or field carcinogenesis) could alter miRNA profiles systemically.

The study’s miRNAs may not be "cancer biomarkers" per se, but indicators of a pro-tumorigenic systemic environment. This could still be clinically useful for risk stratification but requires mechanistic validation.

  • Timing of Blood Sample Collection:
    1. The study used pre-diagnostic blood samples collected at baseline (recruitment), with an average follow-up of 9.6 years until cancer diagnosis. This long interval might explain the lack of significant miRNA differences, as molecular changes associated with cancer may become detectable only closer to clinical diagnosis or during later stages of tumor progression.
    2. For miRNAs like miR-21, which are linked to advanced stages, therapy resistance, and poor prognosis, their expression might not be altered enough in the very early phases of carcinogenesis to be detected years before diagnosis.
  • Methodological Rigor:
    1. The study design is robust, with prospective sample collection, careful matching of cases and controls, and adjustment for confounders. The use of Bonferroni correction, while conservative, reduces the risk of false positives.
    2. The lack of significant findings after multiple testing correction suggests that the observed associations (e.g., Let-7, miR-222) may be marginal or require larger sample sizes for validation.
  • Biological Plausibility:
    1. The miRNAs studied (e.g., Let-7, miR-21, miR-222) have well-documented roles in cancer, but their utility as early biomarkers may depend on the temporal dynamics of their expression during carcinogenesis. The study’s negative results could reflect the "too early" sampling issue rather than a true lack of association.

OBSERVATIONS BY PARAGRAPH:

Minor edits could significantly elevate the paper’s impact.

  1. INTRODUCTION

The Introduction section of the article is too concise and generic.

While their systematic review (Dansero et al., 2022) is a valid starting point, the authors fail to provide a detailed justification of the selection of the specific 8 miRNAs, with deeper biological and clinical context. The authors could strengthen the narrative by citing key studies beyond their own meta-analysis and clarifying the rationale for studying circulating (vs. tissue) miRNAs in a pre-diagnostic setting.

I would suggest the authors add a table summarizing the 8 miRNAs’ roles in CRC to improve clarity.

  1. MATERIAL AND METHODS

2.2. Blood samples were collected at baseline (= recruitment of participants), and the average follow-up duration until the occurrence of colon cancer among cases was 9.6 years. (lines 86-90). For a better interpretation of results, would the authors indicate the full range of occurrence (min-max time of occurrence)?

  1. RESULTS

Stratified Analysis by Time-to-Diagnosis:

If possible, due to the reduced cohort size, the dataset would be analyzed by stratifying cases based on time between blood draw and diagnosis (e.g., <5 years vs. ≥5 years) to uncover stronger associations in subgroups closer to diagnosis, where miRNA dysregulation could be more pronounced.

  1. DISCUSSION -> 4 DISCUSSION

Discussion represents the fourth paragraph: change the numbering.

In this section, while discussing the limitations of their study, the authors should also provide suggestions for improvement.

The authors conclude that “a major limitation of our study is the relatively small sample size, which does not allow us to confirm whether the suggestive associations we observed are true signals” (lines 255-266).

The sample size (104 cases/controls) may be underpowered to detect subtle miRNA differences. Certainly, expanding the sample size by including other cohorts could increase statistical power.

Longitudinal Sampling: Capturing Dynamic Changes

However, as the authors refer to a “prospective epidemiological cohort”, I believe that a longitudinal design with serial blood samples collected at multiple time points (e.g., every 2–3 years) would better capture dynamic changes in miRNA levels as participants approach cancer diagnosis. This approach would validate the potential diagnostic role of certain miRNAs by revealing whether they exhibit gradual upregulation or downregulation during the latent phase of tumor development, even if the pre-diagnosis analysis showed only borderline differences.

Cancer development is a multi-year process involving gradual molecular alterations. By analyzing serial samples, the authors could track how miRNA levels evolve over time in individuals who later develop cancer versus those who remain healthy.

For example, miRNAs like miR-21 (associated with tumor progression) might show a steady increase in cases years before diagnosis, while controls exhibit stable levels. This temporal pattern would strengthen the biomarker’s predictive value.

REFERENCES

While expanding the introduction, some relevant literature would be added.

To that, I would suggest a recently published review:

Federica Longo, Giuseppe Gattuso, Graziana Spoto, Daria Ricci, Anastasia Cristina Venera Vitale, Alessandro Lavoro, Saverio Candido, Massimo Libra, Luca Falzone,

The multifaceted role of microRNAs in colorectal cancer: pathogenesis and therapeutic implications,

Non-coding RNA Research, Volume 14, 2025, Pages 65-95, ISSN 2468-0540,

https://doi.org/10.1016/j.ncrna.2025.05.012.

CONCLUSION:

The study is conceptually original and methodologically rigorous, employing a well-designed nested case-control approach within the EPIC-Italy cohort. However, I have some concerns regarding the interpretation and framing of the results, which I believe should be addressed to strengthen the paper’s impact.

While the negative findings are not necessarily a flaw—given the challenges of identifying pre-diagnostic biomarkers—the manuscript would benefit from a more thorough discussion of the study rationale and escape strategies for future research.

Author Response

Reviewer 1

KEY OBSERVATIONS:

Most "early" miRNA studies in blood are still conducted within 1–2 years of diagnosis, not decades prior. While tissue-based studies implicate these miRNAs in CRC development, their detectability in circulation years before diagnosis remains controversial.

If blood samples were collected years before cancer diagnosis, the detected miRNAs likely reflect indirect systemic changes (e.g., immune/inflammatory responses) rather than direct tumor secretion. Tumor-derived miRNAs may be diluted by systemic noise until late stages. Actively or passively, tumor-released miRNAs (cell death) may be undetectable or negligible in very early stages when the tumor is microscopic or pre-malignant. Early miRNA changes may be tissue-specific but not yet detectable in blood.

Dysregulation of tumor suppressor miRNAs like let7 (silenced in early adenomas) might occur locally in the colonic epithelium long before secretion into circulation reaches detectable levels. Circulating miRNAs could become measurable only after the tumor reaches a certain size, invades vasculature, or triggers systemic inflammation. This is particularly sustainable for miR-21, the most frequently upregulated oncomiR in CRC, associated with advanced-stage disease and poor prognosis.

Conversely, inflammation-driven miRNAs could emerge earlier as paraneoplastic signals: chronic inflammation, metabolic stress, or immune dysregulation (e.g., from nascent tumors or field carcinogenesis) could alter miRNA profiles systemically.

The study’s miRNAs may not be "cancer biomarkers" per se, but indicators of a pro-tumorigenic systemic environment. This could still be clinically useful for risk stratification but requires mechanistic validation.

We thank the reviewer for this insightful interpretation, which fully reflects our own view and the rationale for selecting the miRNAs analyzed in this study. We agree that the association observed years before colon cancer diagnosis could be driven by pre‑tumoral conditions or paraneoplastic signals, such as chronic inflammation, metabolic stress, or early immune dysregulation, which may systemically alter circulating miRNA profiles. We have now emphasized this point more clearly in the revised manuscript.

Timing of Blood Sample Collection:

The study used pre-diagnostic blood samples collected at baseline (recruitment), with an average follow-up of 9.6 years until cancer diagnosis. This long interval might explain the lack of significant miRNA differences, as molecular changes associated with cancer may become detectable only closer to clinical diagnosis or during later stages of tumor progression.

For miRNAs like miR-21, which are linked to advanced stages, therapy resistance, and poor prognosis, their expression might not be altered enough in the very early phases of carcinogenesis to be detected years before diagnosis.

Biological Plausibility:

The miRNAs studied (e.g., Let-7, miR-21, miR-222) have well-documented roles in cancer, but their utility as early biomarkers may depend on the temporal dynamics of their expression during carcinogenesis. The study’s negative results could reflect the "too early" sampling issue rather than a true lack of association.

We fully agree with the reviewer’s interpretation, and we have now incorporated these important considerations into the revised Discussion to better emphasize their relevance to our findings.

Minor edits could significantly elevate the paper’s impact.

INTRODUCTION

The Introduction section of the article is too concise and generic.

While their systematic review (Dansero et al., 2022) is a valid starting point, the authors fail to provide a detailed justification of the selection of the specific 8 miRNAs, with deeper biological and clinical context. The authors could strengthen the narrative by citing key studies beyond their own meta-analysis and clarifying the rationale for studying circulating (vs. tissue) miRNAs in a pre-diagnostic setting.

I would suggest the authors add a table summarizing the 8 miRNAs’ roles in CRC to improve clarity.

In the revised manuscript, we have expanded the Introduction to provide a clearer justification for the selection of the eight miRNAs analyzed in our study, including a biological and clinical context supported by key literature beyond our own meta‑analysis. Furthermore, as recommended, we have added a new table summarizing the roles of these miRNAs in colorectal cancer, their main molecular targets and pathways, and the corresponding references, to improve clarity for the reader.

MATERIAL AND METHODS

2.2. Blood samples were collected at baseline (= recruitment of participants), and the average follow-up duration until the occurrence of colon cancer among cases was 9.6 years. (lines 86-90). For a better interpretation of results, would the authors indicate the full range of occurrence (min-max time of occurrence)?

We have added the full range of time to diagnosis (min–max) in the revised manuscript as requested.

RESULTS

Stratified Analysis by Time-to-Diagnosis:

If possible, due to the reduced cohort size, the dataset would be analyzed by stratifying cases based on time between blood draw and diagnosis (e.g., <5 years vs. ≥5 years) to uncover stronger associations in subgroups closer to diagnosis, where miRNA dysregulation could be more pronounced.

We appreciate the reviewer’s insightful suggestion to stratify cases based on the time interval between blood draw and diagnosis to potentially identify stronger associations closer to diagnosis. We performed such an analysis by dividing cases into two groups: early diagnosis (<5 years) and late diagnosis (≥5 years). However, no significant differences in miRNA associations were observed between these subgroups. This is most likely due to the limited sample size, which reduces the statistical power to detect subgroup-specific effects.

DISCUSSION

Discussion represents the fourth paragraph: change the numbering.

As requested by the Editor during the pre‑submission stage, we structured the manuscript with the Discussion as the third section and the Materials and Methods at the end. We followed this recommendation to ensure consistency with the journal’s preferred format, even though this layout differs from the standard format described in the author guidelines.

In this section, while discussing the limitations of their study, the authors should also provide suggestions for improvement.

The authors conclude that “a major limitation of our study is the relatively small sample size, which does not allow us to confirm whether the suggestive associations we observed are true signals” (lines 255-266).

The sample size (104 cases/controls) may be underpowered to detect subtle miRNA differences. Certainly, expanding the sample size by including other cohorts could increase statistical power.

Longitudinal Sampling: Capturing Dynamic Changes

However, as the authors refer to a “prospective epidemiological cohort”, I believe that a longitudinal design with serial blood samples collected at multiple time points (e.g., every 2–3 years) would better capture dynamic changes in miRNA levels as participants approach cancer diagnosis. This approach would validate the potential diagnostic role of certain miRNAs by revealing whether they exhibit gradual upregulation or downregulation during the latent phase of tumor development, even if the pre-diagnosis analysis showed only borderline differences.

Cancer development is a multi-year process involving gradual molecular alterations. By analyzing serial samples, the authors could track how miRNA levels evolve over time in individuals who later develop cancer versus those who remain healthy.

For example, miRNAs like miR-21 (associated with tumor progression) might show a steady increase in cases years before diagnosis, while controls exhibit stable levels. This temporal pattern would strengthen the biomarker’s predictive value.

By “prospective,” we are referring to the epidemiological definition of a cohort study, in which participants are recruited while healthy, a blood sample is collected at baseline, and they are then followed over time to identify incident cases of disease. The design is therefore prospective with respect to case ascertainment, but not for repeated collection of biological samples.

We fully agree that an ideal study would include serial blood samples collected at multiple time points to identify the moment when circulating miRNAs begin to be released in cases approaching diagnosis. Unfortunately, our study design does not allow for this type of longitudinal analysis. However, we have now included the idea of a follow‑up study with repeated sampling in the Discussion as a valuable direction for future research.

REFERENCES

While expanding the introduction, some relevant literature would be added. To that, I would suggest a recently published review: Federica Longo, Giuseppe Gattuso, Graziana Spoto, Daria Ricci, Anastasia Cristina Venera Vitale, Alessandro Lavoro, Saverio Candido, Massimo Libra, Luca Falzone. The multifaceted role of microRNAs in colorectal cancer: pathogenesis and therapeutic implications. Non-coding RNA Research, Volume 14, 2025, Pages 65-95, ISSN 2468-0540, https://doi.org/10.1016/j.ncrna.2025.05.012.

We greatly appreciate this excellent suggestion. This recent review has been particularly valuable in providing relevant context and has helped us strengthen the background of our manuscript.

REVIEWER 2

Comments and Suggestions for Authors

The manuscript by Padroni et al. entitled “Circulating microRNAs as Early Biomarkers of Colon Cancer: A Nested Case-Control Study within a Prospective Cohort” describes analysis of miRNAs previously associated with colon cancer in the European Prospective Investigation into Cancer and Nutrition (EPIC)-Italy cohort. Although the presented data is not conclusive, it provides valuable insights that can guide future, larger-scale studies. The manuscript is also worth publishing due to application of a comprehensive statistical approach.

However, some revisions are necessary before it can be accepted for publication. Specific remarks are given below:

  • Page 4, line 150: A reference should be given to R Statistical Software and R Studio.

We have added the appropriate references to R Statistical Software and RStudio in line 150 of the revised manuscript.

  • Page 4, lines 155-156: “The mean age at the recruitment is slightly higher among cases than controls (55.8 versus 54.3 years, p=0.058).” – this sentence is inconsistent with Table 1 where mean age in controls is 54.5 years and p-value=0.09. Please double-check and correct.

Thank you for pointing this out. The inconsistency has been corrected, and the values now match Table 1 in the revised manuscript.

  • Description of Y-axes in Figure 1 should be more informative. Does Y-axes show expression normalized to miR-484? Additionally, the figure caption should specify whether the box plots display median or mean values, and whether they include standard deviation (SD), standard error (SE), or other statistical measures.

The Y‑axes in Figure 1 have been clarified to indicate that the expression values are normalized to miR‑484. In addition, the figure caption now specifies that the box plots display median values with interquartile ranges.

  • It seems that for some miRNAs there are errors in the calculation of Fold Changes in Table 2. E.g. for Mir92 mean expression in cases is 44.98, mean expression in controls is 0.68 and the fold change is 1.79. Please double check. Moreover, data presented in Table 2 is not consistent with Figure 1 e.g. for MirLet7 Table 2 shows that expression in controls is lower than in cases, whereas Figure 1 shows that it is higher in controls than in cases. Please check and correct.

We are very grateful to the reviewer for noticing these discrepancies in the results. The fold change values in Table 2 and the corresponding data in Figure 1 have been carefully re‑checked, and the inconsistencies have been corrected in the revised version of the manuscript.

  • Page 7, line 194: The phrase “In conclusion” is not appropriate in this context and should be replaced with a more suitable alternative.

We have removed the wording “In conclusion” while keeping the sentence itself, as suggested, in the revised manuscript.

  • Page 7, lines 198-199: “…and Let7 (OR = 0.91, IC 95%: 0.82–1.00, p = 0.06) showed a significant association with colon cancer status.” To be precise, p=0.06 is not statistically significant if significance level is set at p=0.05.

This is absolutely correct, and we have modified the sentence in the revised manuscript to clarify that the association is not statistically significant at the 0.05 level.

  • Page 9, lines 257-258: “In our study, when we adjusted for major potential confounders, even Let7, which was statistically significant in the univariate models, lost its significance.” It should be noted that it is exactly opposite for Mir222 which is not significant in univariate model but it is significant in multivariate model.

This is correct. We have added a sentence in the Discussion highlighting the opposite behavior of miR‑222, which reaches statistical significance only after adjustment for potential confounders. Thank you for this helpful suggestion.

Reviewer 2 Report

Comments and Suggestions for Authors

The manuscript by Padroni et al. entitled “Circulating microRNAs as Early Biomarkers of Colon Cancer: A Nested Case-Control Study within a Prospective Cohort” describes analysis of miRNAs previously associated with colon cancer in the European Prospective Investigation into Cancer and Nutrition (EPIC)-Italy cohort. Although the presented data is not conclusive, it provides valuable insights that can guide future, larger-scale studies. The manuscript is also worth publishing due to application of a comprehensive statistical approach. However, some revisions are necessary before it can be accepted for publication. Specific remarks are given below:

1) Page 4, line 150: A reference should be given to R Statistical Software and R Studio.

2) Page 4, lines 155-156: “The mean age at the recruitment is slightly higher among cases than controls (55.8 versus 54.3 years, p=0.058).” – this sentence is inconsistent with Table 1 where mean age in controls is 54.5 years and p-value=0.09. Please double-check and correct.

3) Description of Y-axes in Figure 1 should be more informative. Does Y-axes show expression normalized to miR-484? Additionally, the figure caption should specify whether the box plots display median or mean values, and whether they include standard deviation (SD), standard error (SE), or other statistical measures.

4) It seems that for some miRNAs there are errors in the calculation of Fold Changes in Table 2. E.g. for Mir92 mean expression in cases is 44.98, mean expression in controls is 0.68 and the fold change is 1.79. Please double check. Moreover, data presented in Table 2 is not consistent with Figure 1 e.g. for MirLet7 Table 2 shows that expression in controls is lower than in cases, whereas Figure 1 shows that it is higher in controls than in cases. Please check and correct.

5) Page 7, line 194: The phrase “In conclusion” is not appropriate in this context and should be replaced with a more suitable alternative.

6) Page 7, lines 198-199: “…and Let7 (OR = 0.91, IC 95%: 0.82–1.00, p = 0.06) showed a significant association with colon cancer status.” To be precise, p=0.06 is not statistically significant if significance level is set at p=0.05.

7) Page 9, lines 257-258: “In our study, when we adjusted for major potential confounders, even Let7, which was statistically significant in the univariate models, lost its significance.” It should be noted that it is exactly opposite for Mir222 which is not significant in univariate model but it is significant in multivariate model.

Author Response

REVIEWER 2

Comments and Suggestions for Authors

The manuscript by Padroni et al. entitled “Circulating microRNAs as Early Biomarkers of Colon Cancer: A Nested Case-Control Study within a Prospective Cohort” describes analysis of miRNAs previously associated with colon cancer in the European Prospective Investigation into Cancer and Nutrition (EPIC)-Italy cohort. Although the presented data is not conclusive, it provides valuable insights that can guide future, larger-scale studies. The manuscript is also worth publishing due to application of a comprehensive statistical approach.

However, some revisions are necessary before it can be accepted for publication. Specific remarks are given below:

  • Page 4, line 150: A reference should be given to R Statistical Software and R Studio.

We have added the appropriate references to R Statistical Software and RStudio in line 150 of the revised manuscript.

  • Page 4, lines 155-156: “The mean age at the recruitment is slightly higher among cases than controls (55.8 versus 54.3 years, p=0.058).” – this sentence is inconsistent with Table 1 where mean age in controls is 54.5 years and p-value=0.09. Please double-check and correct.

Thank you for pointing this out. The inconsistency has been corrected, and the values now match Table 1 in the revised manuscript.

  • Description of Y-axes in Figure 1 should be more informative. Does Y-axes show expression normalized to miR-484? Additionally, the figure caption should specify whether the box plots display median or mean values, and whether they include standard deviation (SD), standard error (SE), or other statistical measures.

The Y‑axes in Figure 1 have been clarified to indicate that the expression values are normalized to miR‑484. In addition, the figure caption now specifies that the box plots display median values with interquartile ranges.

  • It seems that for some miRNAs there are errors in the calculation of Fold Changes in Table 2. E.g. for Mir92 mean expression in cases is 44.98, mean expression in controls is 0.68 and the fold change is 1.79. Please double check. Moreover, data presented in Table 2 is not consistent with Figure 1 e.g. for MirLet7 Table 2 shows that expression in controls is lower than in cases, whereas Figure 1 shows that it is higher in controls than in cases. Please check and correct.

We are very grateful to the reviewer for noticing these discrepancies in the results. The fold change values in Table 2 and the corresponding data in Figure 1 have been carefully re‑checked, and the inconsistencies have been corrected in the revised version of the manuscript.

  • Page 7, line 194: The phrase “In conclusion” is not appropriate in this context and should be replaced with a more suitable alternative.

We have removed the wording “In conclusion” while keeping the sentence itself, as suggested, in the revised manuscript.

  • Page 7, lines 198-199: “…and Let7 (OR = 0.91, IC 95%: 0.82–1.00, p = 0.06) showed a significant association with colon cancer status.” To be precise, p=0.06 is not statistically significant if significance level is set at p=0.05.

This is absolutely correct, and we have modified the sentence in the revised manuscript to clarify that the association is not statistically significant at the 0.05 level.

  • Page 9, lines 257-258: “In our study, when we adjusted for major potential confounders, even Let7, which was statistically significant in the univariate models, lost its significance.” It should be noted that it is exactly opposite for Mir222 which is not significant in univariate model but it is significant in multivariate model.

This is correct. We have added a sentence in the Discussion highlighting the opposite behavior of miR‑222, which reaches statistical significance only after adjustment for potential confounders. Thank you for this helpful suggestion.

Round 2

Reviewer 1 Report

Comments and Suggestions for Authors

In this article, the authors analyze the expression levels of a panel of eight miRNAs, selected based on their role in colon cancer, in a cohort of 104 individuals who developed cancer within ten years of blood sampling (collected at study enrollment) and 104 healthy controls.

In the revised version, the study is more clearly presented, with better-defined premises and results. Despite the lack of conclusive findings, the research is methodologically sound and well-conducted. The main limitations concern two aspects:

The sample size, which may be insufficient to detect significant differences.

The a priori selection of miRNAs, which could introduce bias.

The study’s objective is ambitious, and to increase the chances of success, an unbiased analysis on a larger cohort—examining the entire miRNome rather than a predefined subset—would be advisable. Overall, this work serves as a valid pilot study for future research.

Author Response

Dear Reviewer,

We thank you for your further review of our manuscript and for the helpful suggestions. We have addressed all comments in the revised version, with changes highlighted in the text.

Please find below our point-by-point responses.

In the revised version, the study is more clearly presented, with better-defined premises and results. Despite the lack of conclusive findings, the research is methodologically sound and well-conducted.

The main limitations concern two aspects:

The sample size, which may be insufficient to detect significant differences.

The a priori selection of miRNAs, which could introduce bias.

The study’s objective is ambitious, and to increase the chances of success, an unbiased analysis on a larger cohort—examining the entire miRNome rather than a predefined subset—would be advisable. Overall, this work serves as a valid pilot study for future research.

We thank the Reviewer for the positive evaluation of our revised manuscript and for acknowledging the methodological soundness of our work. We fully agree with the Reviewer that the limited sample size and the a priori selection of miRNAs represent important constraints that may have affected the strength of our findings. Unfortunately, these limitations could not be overcome in the context of the present study due to resource constraints and the specific design of the project. For this reason, we have explicitly discussed these issues in the revised version of the Discussion section, so that readers are aware of them and can interpret our results accordingly.

Reviewer 2 Report

Comments and Suggestions for Authors

I still see some problems with the manuscript precluding its publication:

1) The following comment is still valid:

“The mean age at the recruitment is slightly higher among cases than controls (55.8 versus 54.3 years, p=0.058).” – this sentence is inconsistent with Table 1 where mean age in controls is 54.5 years and p-value=0.09.

The authors claim that they have corrected the manuscript accordingly, but they have not.

2) Data presented in Figure 1 is still inconsistent with data presented in Table 3. I understand that Table 3 presents mean values while Figure 1 presents median values and IQR, but looking at Figure 1 I cannot believe that mean expression of let7 is four times lower in cases comparing to controls. The same applies to other miRNAs. The authors should provide the raw data so their calculations can be checked.

Author Response

Dear Reviewer,

We thank you for your further review of our manuscript and for the helpful suggestions. We have addressed all comments in the revised version, with changes highlighted in the text.

Please find below our point-by-point responses.

I still see some problems with the manuscript precluding its publication:

1) The following comment is still valid:

“The mean age at the recruitment is slightly higher among cases than controls (55.8 versus 54.3 years, p=0.058).” – this sentence is inconsistent with Table 1 where mean age in controls is 54.5 years and p-value=0.09.

The authors claim that they have corrected the manuscript accordingly, but they have not.

We thank the Reviewer for pointing this out. We were under the impression that this sentence had already been corrected in the previous version of the manuscript, but it appears that something went wrong in the editing process. We have now carefully checked the text and ensured that the sentence is fully consistent with Table 1. The corrected version now reads: The mean age at recruitment was slightly higher among cases than controls (55.8 versus 54.5 years, p = 0.09).” We apologize for the oversight.

2) Data presented in Figure 1 is still inconsistent with data presented in Table 3. I understand that Table 3 presents mean values while Figure 1 presents median values and IQR, but looking at Figure 1 I cannot believe that mean expression of let7 is four times lower in cases comparing to controls. The same applies to other miRNAs. The authors should provide the raw data so their calculations can be checked.

We thank the Reviewer for the comment. We would like to clarify that the boxplots shown in Figure 1 are based on log-transformed expression values. This transformation was applied uniformly across all miRNAs to reduce the influence of extreme values (outliers) and to ensure comparability of scale and distribution among the different miRNAs presented.

Regarding the Reviewer’s concern that “mean expression of let-7 is four times lower in cases compared to controls seems implausible based on Figure 1”, we respectfully point out that this impression may result from a misunderstanding between median values (shown in Figure 1) and mean values (reported in Table 3). The boxplots represent the median and interquartile range (IQR) of the log-transformed expression levels, while Table 3 reports the means of the original (untransformed) data. As such, the visual impression from the boxplots cannot be directly compared to the numerical means in the table, especially considering the skewness typically present in miRNA expression data. Log transformation is a common and appropriate approach in this context, as it stabilizes variance and makes distributions more symmetric, which is particularly important for graphical representation and for ensuring accurate visual comparisons across groups.

We have attached a Word file including the corresponding untransformed boxplots, to better illustrate the differences between the raw and log-transformed data distributions. If you believe this figure could help readers better interpret the results, we could include it as a supplementary figure.

Round 3

Reviewer 2 Report

Comments and Suggestions for Authors

1) In the most recent version of the manuscript available for me (v4) the sentence "The mean age at the recruitment is slightly higher among cases than controls (55.8 versus 54.3 years, p=0.058)" is still uncorrected. I believe that this is due to some problems in the editorial process and the sentence will be corrected in the final version of the manuscript.

2) I thank the authors for the clarifications concerning log transformation of data presented in fig 1. It is a good idea to include untransformed boxplots as a supplementary figure. While preparing this supplementary file be sure to include plots for all miRNAs under study, because in the present version one miRNA is missing. Nevertheless, it is always desirable to deposit raw data in an open access repository where it can be available for the reader. I encourage authors to do this.

Author Response

Dear Reviewer,

please find below our point-by-point responses.

1) In the most recent version of the manuscript available for me (v4) the sentence "The mean age at the recruitment is slightly higher among cases than controls (55.8 versus 54.3 years, p=0.058)" is still uncorrected. I believe that this is due to some problems in the editorial process and the sentence will be corrected in the final version of the manuscript.

I believe you may not have access to the latest version of the manuscript. To ensure we are aligned on the same version, I am attaching the most recent file here in the reviewer comments.

2) I thank the authors for the clarifications concerning log transformation of data presented in fig 1. It is a good idea to include untransformed boxplots as a supplementary figure. While preparing this supplementary file be sure to include plots for all miRNAs under study, because in the present version one miRNA is missing. Nevertheless, it is always desirable to deposit raw data in an open access repository where it can be available for the reader. I encourage authors to do this.

We are happy to include the untransformed boxplots as a supplementary figure, and you will find them complete for all miRNAs under study in this version of the manuscript.

We fully agree with the reviewer that depositing raw data in open access repositories is highly desirable and fosters transparency and scientific progress. However, due to the restrictions imposed by our ethical approval, we are not allowed to deposit individual-level data from the EPIC study, even in anonymised form, and can only present results in aggregated format. Nevertheless, we are strongly committed to promoting data sharing and collaborative research. For this reason, we have included a statement at the end of the manuscript encouraging interested researchers to contact the study coordinator to explore the possibility of conducting joint analyses.

Thank you
